# Effects of Oxidative Stress on the Autophagy and Apoptosis of Granulosa Cells in Broody Geese

**DOI:** 10.3390/ijms24032154

**Published:** 2023-01-21

**Authors:** Li’E Hou, Tiantian Gu, Kaiqi Weng, Yu Zhang, Yang Zhang, Guohong Chen, Qi Xu

**Affiliations:** Jiangsu Key Laboratory for Animal Genetic, Breeding, and Molecular Design, Yangzhou University, Yangzhou 225009, China

**Keywords:** goose, broodiness, oxidative stress, apoptosis, autophagy

## Abstract

Broodiness is an unfavorable trait associated with the cessation of egg laying. Studies have found that excessive granulosa cell apoptosis and autophagy occur during goose broodiness. Other studies have also confirmed that oxidative stress is an important cause of apoptosis and autophagy. However, whether oxidative stress occurs during goose broodiness and whether this oxidative stress causes apoptosis and autophagy have not been fully elucidated. In this study, we investigated the effects of oxidative stress on the autophagy and apoptosis of granulosa cells in broody geese. The results showed higher mRNA expression of genes related to antioxidative stress responses (*GPX*, *SOD-1*, *SOD-2*, *COX-2*, *CAT* and *hsp70*) in pre-broody and broody geese than in laying birds. In addition, increased levels of granulosa cell apoptosis and autophagy were observed in pre-broody geese than in laying geese. Additionally, granulosa cells treated with H_2_O_2_ exhibited increased apoptosis and autophagy in vitro, and these effects were responsible for goose granulosa cell death. Moreover, vitamin E treatment effectively protected granulosa cells from H_2_O_2_-induced oxidative stress by inhibiting ROS production. Correspondingly, granulosa cell apoptosis and autophagy were greatly alleviated by vitamin E treatment. Together, our results demonstrated serious oxidative stress and granulosa cell apoptosis and autophagy in broody geese, and oxidative stress promoted apoptosis and autophagy. Vitamin E alleviated the autophagy and apoptosis of granulosa cells by inhibiting oxidative stress.

## 1. Introduction

Broody behavior is an unfavorable trait in commercial poultry production and is associated with the cessation of egg laying [1,2]. Although the persistent selection and artificial incubation of eggs are used in domestic bird species to reduce the incidence of this behavior, broodiness still occurs in most goose breeds. It has been reported that the incidence of broodiness is 100% in some native goose breeds, and broodiness can last up to 120 days [3]. Therefore, strong brooding behavior is an important cause of low egg production in geese. In general, the ovarian follicles of broody fowl are atrophic, and obvious apoptosis of follicular granulosa cells (GCs) has been observed in chickens and broody geese [4,5]. Recently, autophagy of follicular GCs was also observed during broodiness, and GC autophagy might exacerbate follicular atresia to some extent [6,7]. Therefore, GC apoptosis and autophagy may be important factors that lead to follicle atresia in broody fowls.

In vivo studies have confirmed that GC apoptosis and autophagy occur due to oxidative stress by reactive oxygen species (ROS) overproduction. Follicular ROS promote apoptosis, decrease antioxidant defenses and apoptosis-activating protein levels, and exacerbate follicular atresia [8]. Similar evidence has indicated that ROS produced by various sources can accelerate follicular GC apoptosis [9,10]. Furthermore, another type of cell death (autophagy) is often triggered by oxidative damage to GCs during follicular atresia [11,12]. Another major discovery is that ROS induce autophagy in follicular GCs to regulate the broodiness of geese [6]. In contrast, antioxidant supplementation may be effective in controlling the production of ROS, and this approach is still being explored as a potential strategy to overcome reproductive disorders associated with follicular atresia. Vitamin E (VE), curcumin and melatonin have been confirmed to exert protective effects against GC death [13,14,15].

Currently, the use of antioxidants has become a useful therapeutic approach against different livestock and poultry follicular atresia. VE is considered an ideal antioxidant because it can reach different body tissues depending on its lipophilic, anti-inflammatory, antioxidant and cytoprotective activities [16]. Some studies have clarified that VE plays a critical role in follicular development and growth and follicular atresia by controlling proliferation, differentiation and apoptosis [17,18]. Moreover, in recently published journal articles, Ayo, Sezer and Schweigert et al. discussed the autophagic regulatory actions of VE supplementation in the performance of different livestock and poultry species, including rats, hens and reproductive cows [19,20,21].

The interaction between autophagy and apoptosis in cell death is very complex and unpredictable. Both mechanisms are involved in oxidative stress-induced cell death but can either antagonize or synergize with each other, and differences have been observed between different tissues of the same species and between the same tissues of different species. This finding encouraged us to investigate whether oxidative stress causes apoptosis and autophagy in broody geese and how it can be modulated by the antioxidant VE. The results demonstrated that the suppression of oxidative stress plays a critical role in protection against follicular atresia. Thus, these results not only revealed the physiological mechanism underlying brooding behavior in geese but also might provide a plausible treatment strategy for improving the laying performance of geese.

## 2. Results

### 2.1. Morphological and Histological Characteristics of Follicles during Broodiness in Geese

To explore the morphological characteristics of follicles during broodiness in geese, ovaries were harvested from geese during laying, pre-broody and broody periods, the numbers of follicles were counted, and their morphological characteristics were observed. The results showed that the ovaries of the broody geese were markedly atrophied; the diameters of the follicles of the pre-broody and broody birds were only 5.9 cm and 3.8 cm, respectively, whereas that of laying birds was approximately 13.7 cm (Figure 1A). In addition, the total number of follicles was significantly lower in geese in the brooding period than in the laying period (*p* < 0.01) (Figure 1B). Additionally, only small yellow follicles were observed in the broody period. To further determine the histology and characteristics of follicles during broodiness, the microscopic structures of the follicles and GCs were observed. The results showed that the GC layer in the follicles of brooding geese was thinner, and an irregular arrangement of GCs was observed (Figure 1C,D). We then assessed the potential development of GCs during broodiness in geese. Goose GCs were isolated and identified (Figure 2A,B), and the number of GCs was counted. The number of GCs was lower during the brooding period than during the laying period (*p* < 0.01 or *p* < 0.05) (Figure 2C). Collectively, the data suggested that the ovary is markedly atrophied and causes GC death during goose broodiness.

### 2.2. Ovarian Antioxidant Capacity of Broody Geese

To determine whether oxidative stress occurs during goose broodiness, the activities of antioxidant enzymes and glutathione (GSH) and hydrogen peroxide (H_2_O_2_) contents were measured. As expected, we found that the GSH contents, CAT and SOD activities, and total antioxidant capacity (T-AOC) were remarkably higher in the pre-broody and broody birds than in the laying birds (Figure 3A,B). Additionally, higher mRNA expression of genes related to antioxidative stress responses, including *GPX*, *SOD-1*, *SOD-2*, *COX-2*, *CAT* and *hsp70*, was observed in the pre-broody and broody birds (Figure 3C). Furthermore, the protein and mRNA expression levels of *Nrf2* and *Ho-1* were also significantly higher in broody and pre-broody geese than in laying geese (Figure 4A–C). Together, these data indicated that obvious oxidative stress occurs during goose broodiness, which might result in atrophied ovaries and GC death.

### 2.3. Both Apoptosis and Autophagy Promote GC Death during Broodiness in Geese

GC apoptosis was observed in the laying, pre-broody and broody geese, but the degree of apoptosis differed among these geese. The nuclear chromatin of GC clouds was condensed in the laying birds, whereas nuclear chromatin underwent margination and formed vacuolar bodies (apoptosis bodies) in the pre-broody and broody geese (Figure 5A). Additionally, higher mRNA and protein levels of *Caspase-3* and *Caspase-8* were observed in the pre-broody and broody birds than in the laying birds (Figure 5B,C), whereas the opposite trend was found for *Bcl-2* expression. Moreover, the pre-broody birds showed significantly higher transcription levels of *Caspase-3*, *Caspase-8*, *Caspase-9* and *P53* than the laying birds (Figure 5B). We subsequently found that GCs of the pre-broody or broody birds had few cytoplasmic vesicles that exhibited the typical single-membrane structure of autolysosomes, whereas few autophagic vacuoles were observed in the laying birds (Figure 6A). In addition, higher mRNA and protein levels of *LC3* were observed in the broody and pre-broody birds than in the laying birds. We also found that the pre-broody geese had significantly higher expression levels of *ATG12*, *Beclin1*, *P53* and *p62* than the laying birds (Figure 6A–D). Together, these data indicated that apoptosis and autophagy are the two main mechanisms of cell death.

### 2.4. Oxidative Stress Causes Apoptosis and Autophagy in Goose GCs

To verify whether oxidative stress caused the death of goose GCs during broodiness, we analyzed the apoptosis and autophagy of goose GCs exposed to H_2_O_2_ in vitro. The results showed that the activity of GCs was decreased after H_2_O_2_ treatment in a dose-dependent manner, and 100 µM H_2_O_2_ for 12 h was the optimal condition for the establishment of the oxidation stress model in GCs (Figure 7A–D). GCs treated with H_2_O_2_ exhibited significantly decreased cell viability and increased apoptosis and lactate dehydrogenase (LDH) release. To further confirm the oxidative stress-induced enhancement of autophagy, the formation and distribution of autophagosomes in GCs treated with different concentrations of H_2_O_2_ were measured by monodansylcadaverine (MDC) staining. The results showed that exposure to H_2_O_2_ for 12 h increased the numbers of autolysosomes (green puncta) in GCs (Figure 8). These data demonstrated that H_2_O_2_-induced oxidative stress could lead to goose GC apoptosis and autophagy.

### 2.5. Autophagy Exacerbates Oxidative Stress-Induced Apoptosis in Goose Follicle GCs

Subsequently, we identified the role of autophagy in GCs under oxidative stress. GCs were pretreated with 3-methyladenine (3-MA) to block autophagic activity prior to treatment with H_2_O_2_ (Figure 9A). The results showed that the addition of 3-MA significantly reduced the *ATG12*, *Beclin1*, *LC3* and *p62* mRNA levels (Figure 9B). Additionally, GCs treated with H_2_O_2_ and 3-MA exhibited less apoptosis than control GCs (Figure 10A). The cell viability of GCs treated with H_2_O_2_ in the presence of 3-MA was 1.45%, which was significantly higher than that of GCs treated with H_2_O_2_ alone (0.4%) (Figure 10B). In addition, we examined the mRNA expression of apoptosis-related genes in GCs treated with H_2_O_2_ in the presence of 3-MA. Supplementation with 3-MA resulted in lower *Caspase-3* and *Caspase-8* mRNA levels under oxidative stress conditions (Figure 10C). Together, these results confirmed that autophagy plays a critical role in exacerbating oxidative stress-induced apoptosis in goose follicle GCs.

### 2.6. VE Protects GCs from H_2_O_2_-Induced Oxidative Stress

To investigate the putative attenuating effects of antioxidants on H_2_O_2_-treated goose GCs, four different antioxidants (gallic acid, VE, resveratrol and proanthocyanidins) were selected and their antioxidant effects on H_2_O_2_-treated GCs were evaluated (Figure 11A). The results showed that 40 μM VE could protect GC viability from H_2_O_2_-induced oxidative stress, and VE was more effective than gallic acid, resveratrol and proanthocyanidins (Figure 11A). First, the morphology of H_2_O_2_-treated GCs returned to normal after VE treatment (Figure 11B). Second, VE administration alleviated the H_2_O_2_-induced oxidative stress injury of the GC membrane, as determined by LDH release assay (Figure 11C). Third, VE rescued the decrease in antioxidant capacity by inhibiting ROS production (Figure 12A) and increasing antioxidative-related gene expression in H_2_O_2_-treated GCs (Figure 12B). These data suggested that VE treatment could effectively protect GCs from H_2_O_2_-induced oxidative stress.

### 2.7. VE Alleviates Apoptotic and Autophagic Activities of H_2_O_2_-Treated GCs

We then investigated the effect of VE on the apoptotic and autophagic activities of H_2_O_2_-treated GCs. The results showed that VE rescued the decrease in autophagic activity by reducing the number of autophagic vacuoles and inhibiting the expression of autophagy-related genes (*ATG12*, *Beclin1*, *LC3* and *p62*) in H_2_O_2_-treated GCs (Figure 13A,B). Moreover, VE alleviated H_2_O_2_-induced GC apoptosis in the ovaries by decreasing the expression of apoptosis-related genes (*Caspase-3*, *Caspase-8*, etc.) and increasing the expression of an anti-apoptotic gene (*Bcl-2*) (Figure 14A,B). Collectively, these data suggested that VE treatment could effectively alleviate apoptotic and autophagic activities in H_2_O_2_-treated GCs to maintain homeostasis.

## 3. Discussion

Unlike those of mammals, goose follicles do not undergo atresia. For this reason, geese exhibit long-term continuous egg production, which is influenced by ovarian follicle development and ovulation. Although Zhedong white goose (Anser cygnoides) is one of the most important domesticated avian species and of high economic value in China, its strong broodiness and poor egg-laying performance limit its economic value in the farming industry. The broody behavior of geese is a significant aspect of their reproduction and influences egg production through the degeneration of follicles. Mounting evidence shows that the development of follicles is associated with reproductive endocrine hormones, including GnRH, PRL, LH, FSH, oxytocin, estradiol and progesterone [7]. Some evidence also suggests that the degeneration of follicles is associated with autophagy, apoptosis and homeostasis imbalance [22]. Oxidative stress, which is induced by an imbalance between ROS production and cellular antioxidant defense capacity, has long been hypothesized to be one of the factors capable of triggering apoptosis and thus follicular atresia, particularly in the presence of stressors such as aging [23,24]. In goose GCs, high-level expression of autophagy- and apoptosis-related genes promoted cell death, indicating the presence of a dysfunctional autophagy/apoptosis system during broody periods. Therefore, oxidative stress-activated cell death has become an important topic that is mostly explained by the interaction between autophagy and apoptosis.

Autophagy is a programmed and self-catabolic cellular mechanism that facilitates cellular homeostasis by eliminating and recycling misfolded proteins or defective organelles [25]. Apoptosis is a process in which the cell shrinks, the chromatin condenses, and the cell divides into pieces called ‘apoptotic bodies’ at the end [26]. Under normal conditions, oxidative stress can increase apoptosis through the activation of caspase-3 (pro-apoptotic signal) and induction of DNA fragmentation [27]. Conversely, it has been reported that cells exposed to a heavy concentration of ROS utilize a protective autophagy mechanism to ensure cell survival [28]. The relationship between autophagy and apoptosis is complex and dependent on the context, and this relationship varies under different cellular and stress conditions. Moreover, autophagy can also be a double-edged sword because abnormally enhanced autophagy can potentially induce apoptosis [29,30,31]. For example, antiapoptotic proteins, such as BCL-2, BCL2L2 and MCL1, can block autophagy in cancer [32,33,34]. However, proapoptotic proteins, such as BAD, PMAIP1 and BBC3, stimulate autophagy [35,36]. Overexpression of AURKA can inhibit apoptosis in human cancer [37,38]. However, the relationships among autophagy, apoptosis and oxidative stress in goose ovarian follicles remain poorly defined. To answer this question, we focused on the relationship between these two types of mechanism by suppressing autophagy with 3-MA and then measuring the change in the apoptosis rate. In the present study, we provide the first demonstration that 3-MA significantly inhibits the number of autophagosomes that are formed and the expression of autophagy-related genes in goose GCs. The accumulation of fluorescein isothiocyanate dots, enhanced cell viability and higher apoptosis-related gene expression in 3-MA-treated cells further confirmed that autophagy exacerbates oxidative stress-induced apoptosis in goose follicle GCs. However, additional research is needed to confirm this finding to understand the specific mechanism underlying this phenomenon.

In our study, we found high levels of oxidative stress in the pre-broody period, and this alteration induced excessive autophagy and apoptosis, which ultimately induced strong GC death in broody geese. In fact, emerging evidence from studies on GCs indicates that autophagy may act predominantly as a predeath pathway, exacerbating cellular damage in response to noxious stimuli [39]. In the present study, we found that phase LC3 levels increased during periods of broodiness. Consistent with the reversal of autophagy, the expression of p62/SQSTM1, which is a polyubiquitin-binding protein that is degraded during autophagy, exhibited an increasing trend. These data may prove that oxidative stress leads to aberrant accumulation of the autophagy adaptor p62/SQSTM1, increased intracellular stress and nonprogrammed GC death. Seillier et al. [40] also found that TP53INP1 expression is strongly induced by high oxidative stress and that TP53INP1 interacts with LC3 in autophagosomes, displacing p62 and inducing autophagy-dependent cell death. Whether a protein interacts with LC3 and displaces p62 from autophagosomes during goose broodiness remains to be determined. We will subsequently identify putative proteins, such as TP53INP1, and verify the mechanism through which they interact with LC3 and p62/SQSTM1.

Oxidative stress has been described as a prime driver of GC death during follicular atresia. Gallic acid, VE, resveratrol and proanthocyanidin are well-known antioxidants and protect against oxidative stress. Here, we first proposed that the inhibition of autophagy and apoptosis through novel regulators contributes to VE-mediated GC survival under conditions of oxidative stress. The H_2_O_2_-induced loss of GC viability was significantly reduced after VE administration, and this effect was correlated with the attenuation of autophagic and apoptosis signals upon oxidative stimulation. Interestingly, among the four antioxidants, VE showed the highest activity, and elevated levels of oxidative stress and apoptotic and autophagy biomarkers were reduced by treatment with VE. The ovary is a metabolically active organ that generates excess peroxyl during the final stages of follicular development and ovulation [41,42]. Aneta Baj [43] and colleagues showed that the high antioxidant activity of VE is mainly attributed to stereoelectronic effects exerted by the oxygen atom in the dihydropyran ring. An increasing number of studies have shown that based on kinetic data and the physiological molar ratio of VE to substrates, peroxyl radicals are the only radicals that VE can scavenge efficiently in vivo. Our experimental results also confirm that VE has a better antioxidant capacity than gallic acid, resveratrol and proanthocyanin.

## 4. Materials and Methods

### 4.1. Ethics Approval and Consent to Participate

All animal experimental protocols used in the present study were approved by the Yangzhou Institutional Animal Committee (permit number: YZUDWSY2020-257, 21 October 2020, Jiangsu Province, China). The procedures were performed in accordance with the Regulations for the Administration of Affairs Concerning Experimental Animals (Yangzhou University, China, 2012) and the Standards for the Administration of Experimental Practices (Jiangsu, China, 2008). The experiment design drawing shows in Appendix A.

### 4.2. Chemicals

MDC (catalog # 30432), hydrogen peroxide (H_2_O_2_; catalog # 323381), collagen (catalog # C7661) and type IV collagenase (catalog # C5138) were obtained from Sigma-Aldrich, Inc. (St. Louis, MO, USA). A reactive oxygen species assay kit (DCFH-DA) was purchased from Beyotime Biotechnology (Shanghai, China). The autophagy inhibitor 3-MA (HY-19312) was purchased from MedChemExpress (Monmouth Junction, NJ, USA). VE, proanthocyanins, resveratrol and gallic acid were purchased from Solarbio (Beijing, China).

### 4.3. Animals and Sample Collection

The animals were obtained from Yangzhou Tiange Goose Industry Co., Ltd. (Yangzhou, Jiangsu, China) and Four Seasons Goose Co., Ltd. (Changzhou, Jiangsu, 377 China). The identification of geese at the early (pre-broody) or middle (broody) stages of broodiness was performed according to the methods described by Yao et al. [3]. Geese (including pre-broody, broody and laying geese) were sacrificed by manual exsanguination immediately after they were anesthetized with sodium pentobarbital, and the ovaries were excised from the left side of the abdominal cavity. After the removal of connective tissues, the ovaries were briefly washed in PBS to remove excess blood and were either immediately fixed in 10% neutral buffered formalin for histological processing or used for the isolation of GCs by follicle puncture.

### 4.4. Hematoxylin-Eosin (HE) Staining

The left ovaries were fixed in 4% formaldehyde for 72 h at room temperature, dehydrated with a series of gradient ethanol solutions, transferred to xylene and embedded in paraffin wax. Three sections with a thickness of 5 mm from each ovary, including the largest cross-section that was cut along the ovarian suspensory ligament and the two adjacent sections on the left and right sides of the largest one, were stained with hematoxylin and eosin and examined under a Nikon 90i microscope (Nikon, Tokyo, Japan). At least three fields of view per section were selected for morphometric analysis (200× magnification). Morphologically normal follicles were categorized into three groups according to their developmental stages: primordial follicles that contained an inner oocyte surrounded by a layer of flattened GCs, primary follicles whose GCs had undergone a progressive transformation from a flattened to a cuboidal shape and secondary follicles with more than two layers of GCs and clearly defined theca cell layers. For each follicular category, the diameters of follicles with well-defined histomorphological characteristics in each field of view were measured using ImageJ software (National Institutes of Health, Bethesda, MD, USA), and the mean diameter of each follicular category was calculated as the mean of all observations for all three individuals of each species.

### 4.5. Cell Culture and Reagents

Laying geese were immediately anesthetized with sodium pentobarbital, and ovaries were harvested from the left side of the abdominal cavity. The ovaries were immediately transferred to the laboratory in phosphate-buffered saline (PBS, HyClone, Logan, UT, USA) supplemented with 2% penicillin–streptomycin (Gibco, Carlsbad, CA, USA) and maintained at 38.5 °C. After six washes with 1% penicillin–streptomycin in sterilized PBS, the connective tissue of the theca membrane was removed with microfine tweezers, the follicle was punctured and the contents were released. Ophthalmic scissors were then used to cut the granular cell layer into 1-mm^3^ pieces and each piece was transferred to a centrifuge tube containing 20 ng/mL type II collagenase (Gibco, CA, USA) solution and digested in a 37 °C water bath for 20 min (shaken every 5 min). After digestion was complete, low-glucose DMEM (HyClone) supplemented with 15% FBS (Gibco) was added to terminate the digestion. The GCs were then filtered with wire on mesh 200 (Solarbio, Beijing, China), centrifuged at 1000× *g* for 5 min and immediately retrieved for cell culture (38.5 °C, 5% CO_2_). The cells were sub-cultured at a ratio of 1:2 once 95–100% confluence had been reached. The second to third passages (P2–3) of GCs were used for the following experiments.

### 4.6. Immunofluorescence

GCs were seeded on 6-well plates at a density of 5 × 10^5^ cells/well and grown to approximately 70% confluence. The cells were then washed two to three times with PBS, fixed in 4% PFA at room temperature (RT) for 30 min and permeabilized with 0.5% Triton-X for 20 min. After washing with PBS, the cells were blocked with 1% BSA (Sigma-Aldrich, St. Louis, MO, USA) for 30 min at RT. The GCs were incubated with anti-FSHR (1:200) (Proteintech, Tokyo, Japan) antibody at 4 °C overnight and then with FITC-labeled goat anti-rabbit IgG (1:200) (Bibang, Yangzhou, China) in the dark for 1 h, and the nuclei were counterstained with DAPI (Bibang, Yangzhou, China) for 5 min at room temperature. Fluorescence images were captured using a laser-scanning confocal fluorescence microscope (Olympus Corporation, Tokyo, Japan, SP70).

### 4.7. Establishment of the Oxidative Stress Model and Antioxidant Treatment

GCs from goose ovaries were seeded onto 96-well plates (1 × 10^4^ cells/well). Once the cells reached 80–90% confluence, they were treated with 0, 50, 100, 150, 200 or 250 μmol/L H_2_O_2_, and the plate was incubated at 38.5 °C in 5% CO_2_ to induce oxidative stress-induced cell death. Twelve hours after treatment with H_2_O_2_, the cells were rinsed with warm PBS, and cell viability was assessed by CCK8 and LDH release assays. Based on a comprehensive assessment of the abovementioned test results, an optimal oxidative stress model was established. We then designed an antioxidant test for differential growth in the presence of optimal oxidative stress (100 μmol/L H_2_O_2_). Four different antioxidants, VE, proanthocyanins, resveratrol and gallic acid, were dissolved in DMSO (a solution of 0, 50, 100, 150, 200 and 250 μmol/L of each antioxidant was prepared) and separately added to GCs. Their antioxidant effects on H_2_O_2_-treated GCs were then evaluated.

### 4.8. Enzyme-Linked Immunosorbent Assay (ELISA)

The CAT activity, SOD activity, GSH content, H_2_O_2_ content and T-AOC in serum and ovary tissues were determined using ELISA kits (JianCheng, Nanjing, China) according to the manufacturer’s instructions. Briefly, 50 μL of serum or homogenized ovary sample was incubated in microtiter wells coated with goat anti-mouse monoclonal antibody against estradiol or progesterone. After incubation and washing, the CAT activity, SOD activity, GSH content, H_2_O_2_ content and T-AOC were determined.

### 4.9. RNA Isolation and Quantitative Real-Time PCR

Total RNA was extracted from goose ovary tissues using TRIzol reagent (Invitrogen, Waltham, MA, USA) following the standard manufacturer’s instructions. The concentration of RNA was determined using a NanoDrop ND-2000 Spectrophotometer (Thermo Scientific™, Waltham, MA, USA, ND-2000). Subsequently, the RNA was reverse-transcribed and amplified using a FastQuant RT Kit (with gDNase), following the standard instructions (Tiangen, Sichuan, China). *GPX*, *SOD-1*, *SOD-2*, *CAT*, *COX-2*, *Hsp70*, *ATG12*, *Beclin-1*, *LC3*, *P62*, *Bcl-2*, *Caspase-3*, *Caspase-8*, *Caspase-9* and *p53* gene sequences were obtained from NCBI, and primers were designed by oligo 7 and synthesized by TSINGKE Biological Company (Beijing, China). PowerUp SYBR Green Master Mix (Applied Biosystems, Waltham, MA, USA, 720733) was used for real-time PCR analysis of the mRNA expression levels of the target genes (Life Technologies, Carlsbad, CA, USA) and the internal control *GAPDH* with a LightCycler 96 Real-Time PCR Detection System (Applied Biosystems, QuantStudio 5, USA). The reactions were performed in a total volume of 10 μL per sample, which included 5 μL of SYBR Green Master Mix, 0.4 μL of forward/reverse primer, 2 μL of diluted cDNA template and 2.2 μL of RNA-free water (Vazyme, Nanjing, China). The cycling conditions were as follows: a holding step at 95 °C for 2 min followed by 40 cycles of 15 s at 95 °C and 1 min at 60 °C. To detect and validate the specific amplification of PCR products, a dissociation curve analysis of the products was conducted at the end of each PCR. Each sample was run in triplicate for analysis. Relative expression levels were calculated using the comparative threshold cycle method and determined by the 2^−ΔΔCt^ method.

### 4.10. Transmission Electron Microscopy

Fresh goose ovary tissues were fixed with 2.5% glutaraldehyde (*v*/*v*) and 2% paraformaldehyde (Solarbio, p1110-500) for 2 h, rinsed with 0.1 M sodium phosphate buffer, dehydrated via a series of gradient ethanol solutions and embedded in epoxy (low-viscosity agar). Ultrathin sections of ovary tissues were collected on epoxy (low-viscosity agar) 100 mesh formvar-coated grids and stained with 1% uranyl acetate and 1% lead citrate. The sections were then viewed with an H600 transmission electron microscope (Hitachi, Tokyo, Japan) at 80 kV.

### 4.11. Immunohistochemistry

The goose ovary tissues were rapidly harvested, fixed with 4% paraformaldehyde and embedded in paraffin for immunohistochemical staining. Tissue sections (thickness of 5 μm) were incubated with primary antibodies against Hsp70, Caspase-3, LC3, p62, Nrf2 and HO-1 overnight at 4 °C and then with secondary antibodies (anti-rabbit IgG, Abcam) at 37 °C for 1 h. The following specific antibodies were used, and the average integrated optical density (IOD) of the cells that positively expressed Hsp70, Caspase-3, LC3, p62, Nrf2 and HO-1 in five randomly selected regions per sample was measured at a magnification of 200× using Image-Pro Plus 5.1 image analysis software.

### 4.12. Cell Proliferation Assay

A cell proliferation test kit was used according to the manufacturer’s instructions (Solarbio, GW770, Beijing, China). GCs from goose ovaries were seeded onto 96-well plates (1 × 10^4^ cells/well). Once the cells reached 70–80% confluence, they were treated with 3% H_2_O_2_ or 3% H_2_O_2_ + VE for 12 h, and a Cell Counting Kit-8 solution was added after the appropriate time (0, 24, 48 or 72 h). Before detection, 10 μL of CCK-8 reagent was added to the culture medium in each well and the plate was incubated at 38.5 °C in a 5% CO_2_ incubator for 4 h. The absorbance at 450 nm was measured with a microplate reader (Infinite M200 Pro, Tecan, Männedorf, Switzerland) and all experiments were performed in triplicate.

### 4.13. Measurement of the Intracellular ROS Levels

The ROS levels in GCs treated with H_2_O_2_ were measured using a Reactive Oxygen Species Assay Kit (Beyotime Biotechnology, Nantong, China). Briefly, the cells were seeded at a density of 1.0 × 10^4^ cells/well on a 96-well plate and then treated with H_2_O_2_. The medium in each well was removed and the cells were then incubated with 10.0 μmol/L 2,7-dichlorodihydrofluorescein diacetate (DCFH-DA), which is easily oxidized to fluorescent dichlorofluorescein (DCF) by intracellular ROS, for 20 min at 38.5 °C in a humidified 5.0% CO_2_ atmosphere. We observed the cells by fluorescence microscopy (Olympus) and measured their fluorescence with excitation and emission wavelengths of 488 and 525 nm, respectively, using a fluorescence spectrophotometer (BioTek, Winooski, VT, USA).

### 4.14. MDC Staining for Autophagic Vacuoles

MDC is a specific autophagy marker that can be absorbed by cells and selectively accumulate in autophagic vesicles. Foci of autophagic vesicles that are labeled with MDC and are distributed in the cytoplasm and around the nucleus can be observed by fluorescence microscopy. Therefore, the induction of autophagy can be determined based on changes in intracellular fluorescent particles [44]. GCs were seeded onto 6-well plates with sterile coverslips and incubated for 24 h. The cells were then treated with 500 µM H_2_O_2_ in the presence or absence of delphinidin chloride, rapamycin or chloroquine for 2 h. After treatment, the cells were washed three times with 1× PBS. After washing, autophagic vacuoles were labeled with MDC by incubating the cells grown on coverslips with 0.05 mM MDC in PBS at 37 °C for 15 min in the dark (protected from direct exposure to light). After incubation, the cells were washed four times with PBS and immediately analyzed with an inverted fluorescence microscope (Leica, Wetzlar, Germany). For the MDC assay, GCs were treated with 100 µM H_2_O_2_ for 12 h. After this incubation period, the cells were incubated with 0.05 mM MDC for 10 min at 38.5 °C and then washed four times with PBS (pH 7.4). The cells were immediately analyzed by fluorescence microscopy and quantified.

### 4.15. Statistical Analysis

All the data were statistically analyzed by one-way analysis of variance using SPSS 20.0 software. Statistically significant results were further analyzed by Duncan’s multiple range test. The data are all presented as the means ± standard errors of the mean. A *p* value < 0.05 was considered to indicate significance. All the figures were generated with GraphPad Prism 5.0.

## 5. Conclusions

Overall, our results demonstrated serious oxidative stress, GC apoptosis and autophagy in broody geese and showed that oxidative stress promoted apoptosis and autophagy. Our study also provides the first line of evidence showing the importance of autophagy- and apoptosis-activated cell death in the attenuation of oxidative stress-induced goose broodiness using VE.

## Figures and Tables

**Figure 1 ijms-24-02154-f001:**
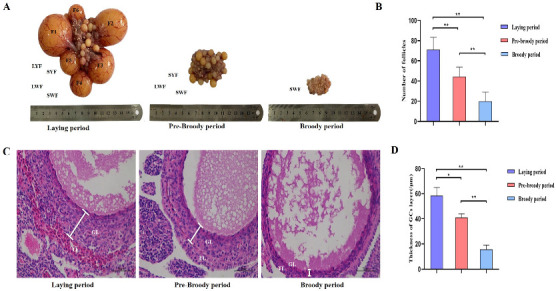
Morphological and histological characteristics of follicles during goose broodiness. (**A**) Ovarian morphology during the process of goose brooding (including the laying, pre-broody and broody periods). F1, F2, F3, F4, F5 and F6 represent the first, second, third, fourth, fifth and sixth yellow hierarchical follicles, respectively. LYF and SYF indicate large and small prehierarchical yellow follicles with diameters < 7 mm, and LWF and SWF respectively represent large and small white follicles with diameters < 5 mm. (**B**) Total number of hierarchical and prehierarchical follicles in goose ovaries (*n* = 5). (**C**) Representative morphology of ovarian LWF follicle tissues demonstrated by H&E staining. TL indicates the theca layer; GL indicates the granulosa layer. Scale bar = 100 μm. (**D**) The diversification of the thicknesses of the GC layer was calculated using ImageJ2x software (*n* = 5). The deviation of 3 areas was randomly selected from each slice for image acquisition, and 3 areas were randomly selected from each image for measurement. * indicates a significant difference (*p* < 0.05) and ** indicates an extremely significant difference (*p* < 0.01).

**Figure 2 ijms-24-02154-f002:**
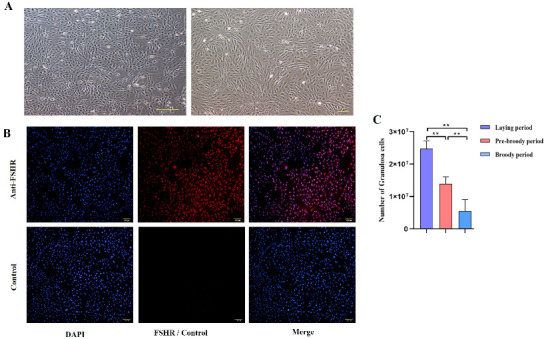
Change in the trend of the GC number. (**A**) Follicular GCs from geese were obtained by collagenase digestion and cultured, and images of goose GCs at 100× and 200× are shown. (**B**) The protein expression of follicle-stimulating hormone receptor (FSHR) in primary GCs was identified by indirect immunofluorescence staining. Left: DAPI staining of the cell nucleus; middle: fluorescence staining for FSHR; right: combined image of DAPI and FSHR staining. (**C**) The number of GCs was calculated by Trypan Blue counting (*n* = 5). All the data are expressed as the means ± standard errors; ** indicates an extremely significant difference (*p* < 0.01).

**Figure 3 ijms-24-02154-f003:**
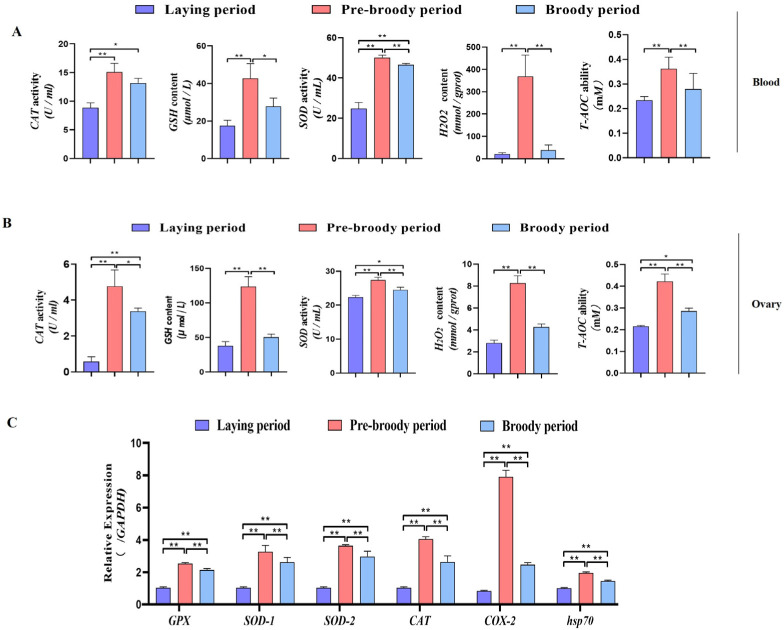
Analysis of oxidation indices of geese in vitro. (**A**) Changes in the CAT, GSH, SOD, H_2_O_2_ and T-AOC levels in blood during different periods (*n* = 8); * *p* < 0.05; ** *p* < 0.01. (**B**) Antioxidant levels in the ovaries of geese during broodiness (*n* = 8). (**C**) Expression of *GPX*, *SOD-1*, *SOD-2*, *COX-2*, *CAT* and *hsp70* in the ovaries of geese during different periods (*n* = 3). The values show the means ± standard errors of all experiments. Asterisks indicate significant differences (* *p* < 0.05, ** *p* < 0.01).

**Figure 4 ijms-24-02154-f004:**
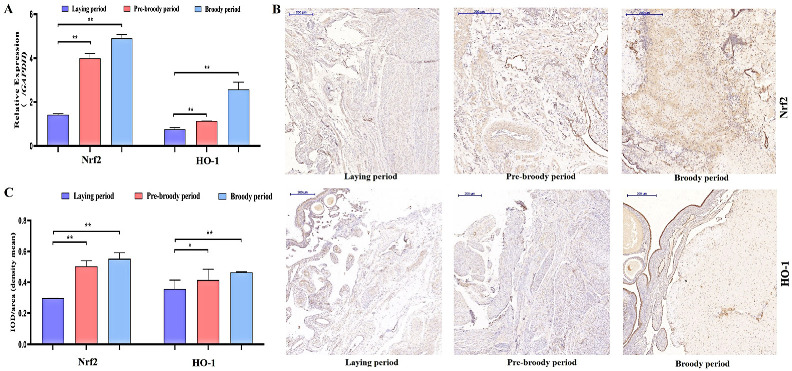
Analysis of oxidation indices of goose ovaries. (**A**) Expression of *Nrf2* and *HO-1* in the ovaries of geese during different periods (*n* = 3). The values show the means ± standard errors of all experiments. Asterisks indicate significant differences (** *p* < 0.01). (**B**) Immunohistochemical staining of Nrf2 and HO-1. Scale bar = 200 μm. (**C**) Quantitative analysis of immunohistochemical staining of Nrf2 and HO-1 based on the mean density shown in (**B**) (*n* = 5); * *p* < 0.05; ** *p* < 0.01.

**Figure 5 ijms-24-02154-f005:**
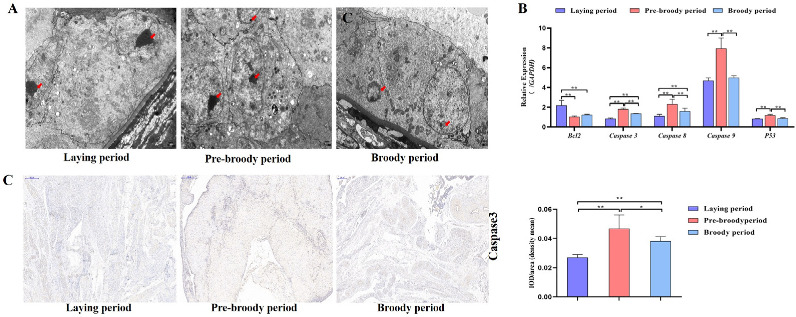
Changes in apoptotic goose ovaries during each period. (**A**) Apoptosis of GCs during the laying, pre-broody and broody periods observed by electron microscopy. Scale bar = 2 μm. Red arrowheads: nuclear chromatin condensation, nuclear chromatin margination. (**B**) RT–qPCR analysis of the mRNA levels of *Bcl-2*, *Caspase-3*, *Caspase-8*, *Caspase-9* and *P53* in the ovaries of geese during different periods. All the data are expressed as the means ± standard deviations of 3 independent experiments and were normalized to *GAPDH*. (*n* = 3); * *p* < 0.05; ** *p* < 0.01. (**C**) Immunohistochemical staining of *Caspase-3* and quantitative analysis of immunohistochemical staining of *Caspase-3* based on the mean density. Scale bar = 200 μm. All the data are expressed as the means ± standard errors of 3 independent experiments and were normalized to *GAPDH* (*n* = 3); * *p* < 0.05; ** *p* < 0.01.

**Figure 6 ijms-24-02154-f006:**
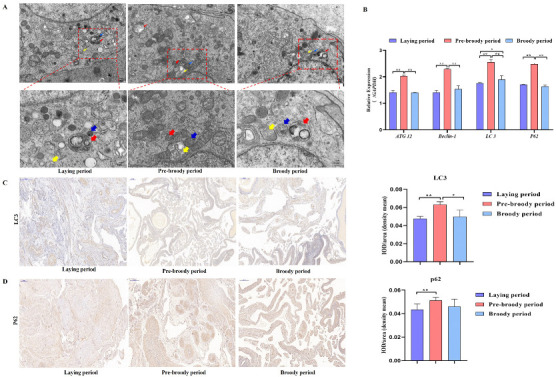
Changes in goose ovary autophagy during each period. (**A**) Autophagy of GCs during the laying, pre-broody and broody periods observed by electron microscopy. The areas indicated by boxes in Upper are shown at higher magnification in Lower. (Yellow arrowheads: autophagosome; Blue arrowheads: lysosome; Red arrowheads: autolysosome; Scale bars: A, Upper, 2.0 μm; A, Lower, 1.0 μm.) (**B**) Relative expression of *ATG12*, *Beclin-1*, *LC3* and *p62* in the ovaries of geese during different periods (*n* = 3). (**C**,**D**) Immunohistochemical staining of *LC3* and *p62*. Scale bar = 200 μm. Quantitative analysis of immunohistochemical staining of *LC3 and p62* based on the mean density. All the data are expressed as the means ± standard errors of 3 independent experiments and were normalized to *GAPDH* (*n* = 3); * *p* < 0.05; ** *p* < 0.01.

**Figure 7 ijms-24-02154-f007:**
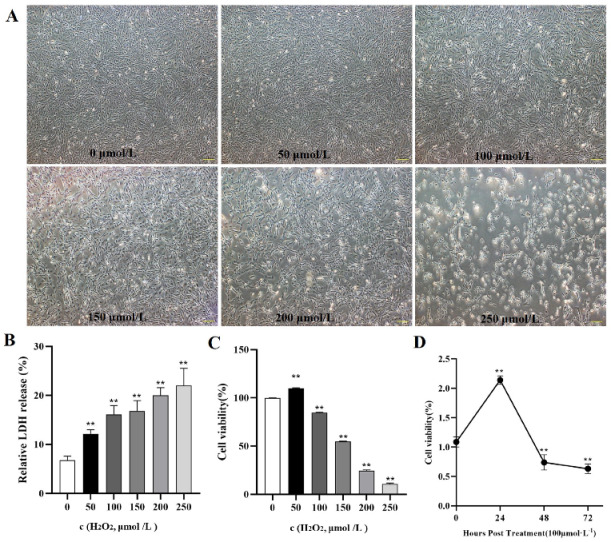
In vitro H_2_O_2−_induced oxidative stress model. (**A**) State and viability of goose GCs treated with different concentrations of H_2_O_2_ (0, 50, 100, 150, 200 and 250 μmol/L). (**B**) Cell death was assessed after H_2_O_2_ treatment for 12 h by LDH release assay. (**C**) After GCs were incubated with various concentrations of H_2_O_2_ for 12 h, the viability of the cells was determined by CCK-8 assay (*n* = 3). (**D**) GCs were incubated with 100 μmol/L H_2_O_2_ for 24, 48 and 72 h, and the viability of the cells was determined by CCK-8 assay (*n* = 3) and compared with that of cells treated with 100 μM H_2_O_2_ for various durations. The values are expressed as the means ± standard errors; ** indicates an extremely significant difference (*p* < 0.01).

**Figure 8 ijms-24-02154-f008:**
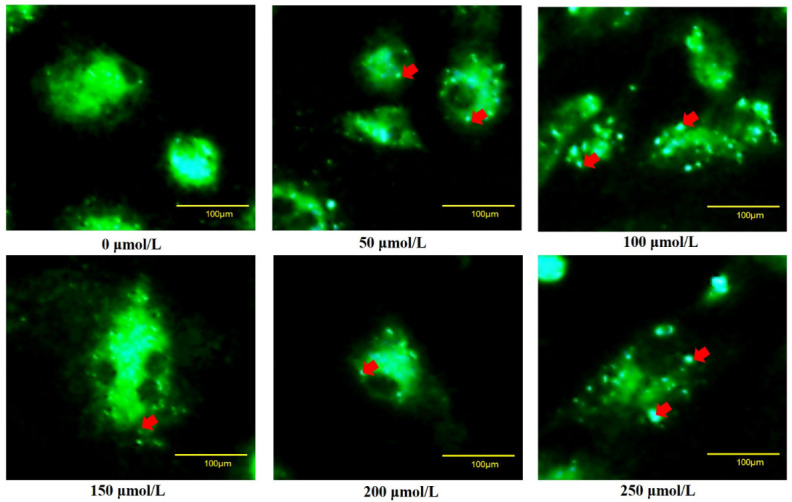
Fluorescence microscopy of MDC staining. GCs transfected with an MDC plasmid were incubated with 100 μM H_2_O_2_ for 12 h. Red arrow: Laser confocal scanning microscopy was used to observe the fluorescent autolysosomes puncta in GCs. Scale bar = 100 μm.

**Figure 9 ijms-24-02154-f009:**
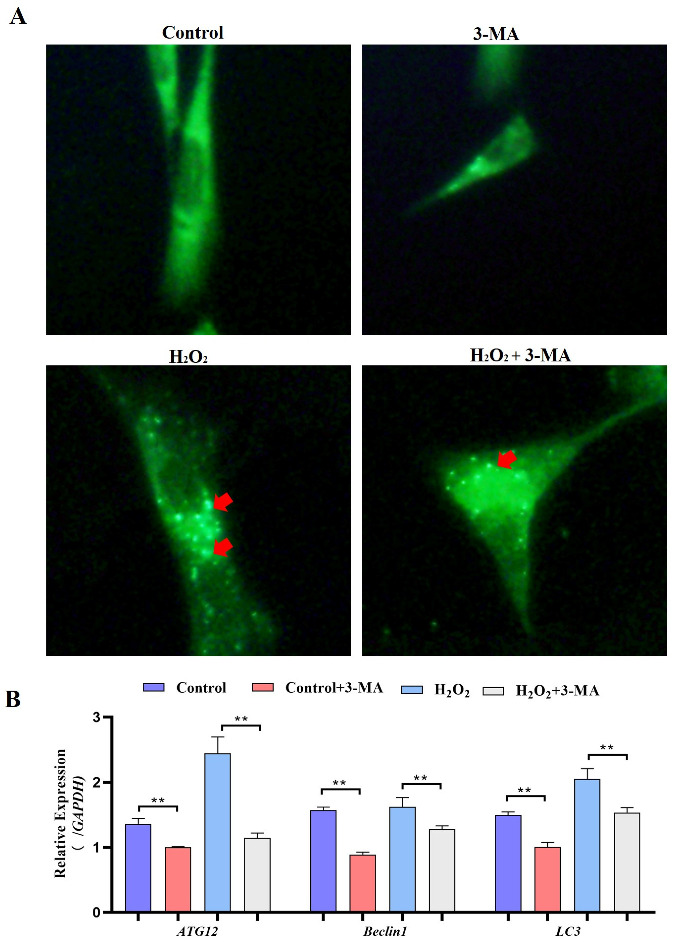
Oxidative stress conditions inhibit autophagy. (**A**) For the MDC assay, GCs were treated with H_2_O_2_ in the presence or absence of 3-MA for 4 h. Following this incubation period, the cells were incubated with 0.05 mM MDC for 10 min at 38.5 °C and then washed 4 times with PBS (pH 7.4). The cells were immediately analyzed by fluorescence microscopy and quantified. Red arrow: Laser confocal scanning microscopy was used to observe the fluorescent autolysosomes puncta in GCs. (**B**) Expression of *Atg12*, *Beclin1*, *LC3* and *p62* after treatment with H_2_O_2_ in the presence or absence of 3-MA for 4 h (*n* = 3). ** indicates an extremely significant difference (*p* < 0.01).

**Figure 10 ijms-24-02154-f010:**
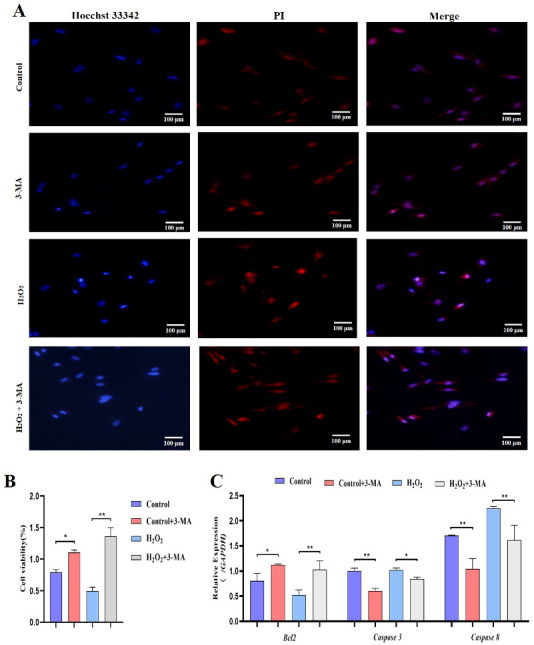
Oxidative stress conditions inhibit autophagy to promote proliferation. (**A**) TUNEL assays were performed using the Promega Dead-end™ Fluorometric TUNEL system kit according to the instructions. (**B**) Cell viability was determined by CCK-8 assay (*n* = 3). (**C**) Expression of *Bcl-2*, *Caspase-3* and *Caspase-8* after treatment with H_2_O_2_ in the presence or absence of 3-MA for 4 h (*n* = 3). The values show the means ± standard deviations of all the experiments. The data represent the means ± standard errors of 3 independent experiments (* *p* < 0.05; ** *p* < 0.01).

**Figure 11 ijms-24-02154-f011:**
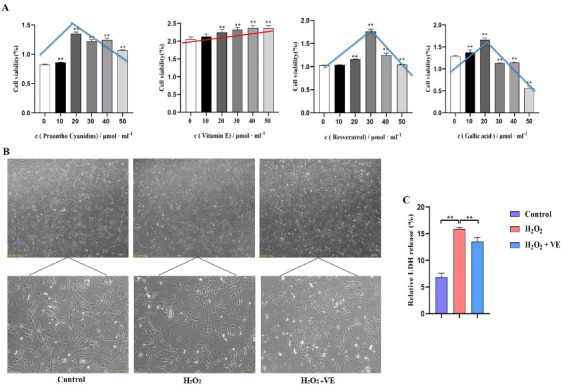
Role of antioxidants in GCs. (**A**) The effects of different antioxidants on the viability of GCs were determined by CCK−8 assay; red/blue lines show the cell viability variation trend with different antioxidants. (**B**) Cell morphology and the state of goose GCs treated with VE under H_2_O_2_—induced oxidative stress conditions. (**C**) The effect of VE on alleviating H_2_O_2_—induced oxidative stress injury of GC membranes was explored by LDH release assay. The data represent the means ± standard errors of 3 independent experiments (** *p* < 0.01).

**Figure 12 ijms-24-02154-f012:**
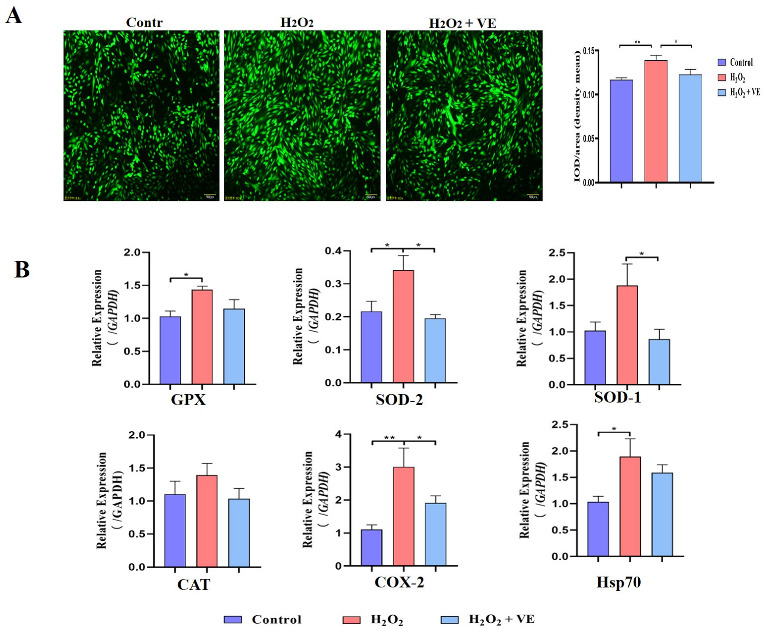
Mechanisms of action of VE antioxidants under H_2_O_2_-induced oxidative stress. (**A**) Fluorescence probe staining of ROS observed by fluorescence microscopy at 488 nm and relative fluorescence intensity of ROS. Left: representative images of ROS staining; right: ROS fluorescence intensity. (**B**) Expression of *GPX*, *SOD-1*, *SOD-2*, *CAT*, *COX-2* and *Hsp70* in VE-treated GCs exposed to H_2_O_2_-induced oxidative stress (*n* = 3). The values are the means ± standard deviations of all the experiments. The data represent the means ± standard errors of 3 independent experiments (* *p* < 0.05; ** *p* < 0.01).

**Figure 13 ijms-24-02154-f013:**
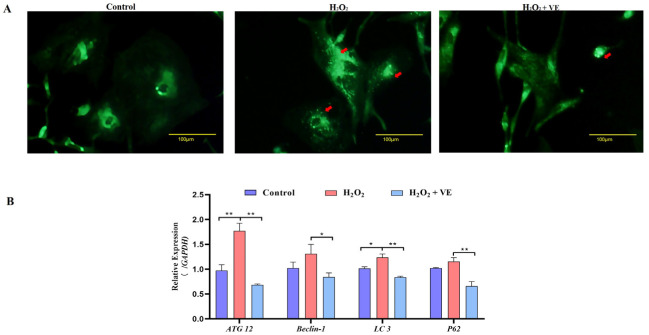
Changes in the levels of autophagy induced by VE antioxidants under H_2_O_2_-induced oxidative stress. (**A**) GCs transfected with an MDC plasmid and subjected to H_2_O_2_-induced oxidative stress were incubated with VE. Laser confocal-scanning microscopy was used to observe fluorescent GFP puncta in the GCs. Scale bar = 100 μm. The cells were incubated with 0.05 mM MDC for 10 min at 38.5 °C and then washed 4 times with PBS (pH 7.4). The cells were immediately analyzed by fluorescence microscopy and quantified. Red arrow: Laser confocal scanning microscopy was used to observe the fluorescent autolysosomes puncta in GCs. (**B**) Expression of *ATG12*, *Beclin1*, *LC3* and *p62* after treatment with VE under H_2_O_2_-induced oxidative stress (*n* = 3). All the data represent the means (± SE) of 3 independent experiments (* *p* < 0.05; ** *p* < 0.01).

**Figure 14 ijms-24-02154-f014:**
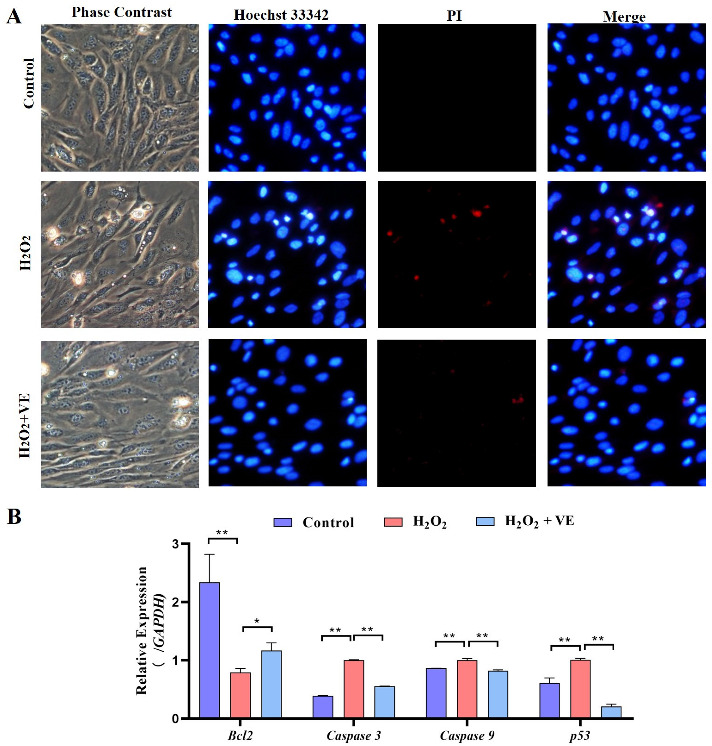
Changes in the levels of apoptosis of VE antioxidants under H_2_O_2_-induced oxidative stress. (**A**) TUNEL assays were performed according to the instructions of the Promega Dead-end™ Fluorometric TUNEL system kit. (**B**) Expression of *Bcl-2*, *Caspase-3*, *Caspase-9* and *p53* after treatment with VE under H_2_O_2_-induced oxidative stress (*n* = 3). * *p* < 0.05; ** *p* < 0.01.

## Data Availability

The data presented in this study are available upon request from the corresponding author.

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
