# Peer review of "Effects of Oxidative Stress on the Autophagy and Apoptosis of Granulosa Cells in Broody Geese"

_ijms, 2023, doi:10.3390/ijms24032154_

Round 1
Author Response
Responses to Reviewer Comments
Point 1: In line 377 “Four Seasons Goose Industry” Are the geese used in the experiment from one season or from the same season?
Response 1: We apologize for our unclear descriptions. ‘Four Seasons Goose Industry’ is the location of the sampling sites, and we have revised the manuscript as follows: “The animals were obtained from Yangzhou Tiange Goose Industry Limited Company (Yangzhou, Jiangsu, China) and Four Seasons Goose Limited Company (Changzhou, Jiangsu, 377 China).” (Lines 406-408)
Point 2: What about the environmental conditions surrounding the geese, such as temperature and humidity, the nature of housing and the feed provided.
Response 2: All geese were provided the same diet, which was combined with coarse and concentrated material and provided ad libitum (Table 1). The geese were kept under a standard light regimen consisting of 14 h of light (14L:10D) during the laying period [1]. The nests were used with rice husk as the mattress, which was sufficiently large (96 cm×85 cm) to comfortably hold several standing geese and allowed the geese to turn around easily.
Table 1. Composition and nutrient content of the experimental diet (air-dry basis).
|
Item |
Content(%) |
|
Corn powder |
40.25 |
|
Crushed wheat |
25 |
|
Crushed barley |
10 |
|
Bean cake powder |
13.25 |
|
Green hay powder |
4 |
|
Limestone |
4.5 |
|
Calcium hydrogen phosphate |
1.5 |
|
Salt |
0.5 |
|
Vitamin and microelement additives |
1 |
|
Total |
100 |
Reference:
[1] Wang, B. W. 2009. Chinese Goose Industry. Shandong Science and Technology Press, Shandong
Point 3: What are the numbers of geese used according to the state of broodiness and egg production, as stated in the research?
Response 3: The number of animals and groups of animals in the study were added to the “Animals and sample collection” subsection of the Materials and Methods section, the supplementary material and the “figure notes” in the Results section in the manuscript.
In our study, we efficiently combined in vitro and in vivo assays. In the in vitro experiment, the number of animals in the experiment group was chosen from the viewpoint of animal welfare. We slaughtered and sampled 5 geese from each group (pre-broody period, laying period and broody period). Moreover, we collected 8 blood samples to observe the oxidative stress levels in each group and used these samples for ELISA. In the in vivo experiment, the GCs analyzed were primary cells isolated from goose ovarian follicles. Therefore, the numbers of geese used was based on the number of actual trials.
We provide a schematic of the experimental design below and in the manuscript. (Lines 578-581; Figure S).
Figure S. Experimental design and workflow for assessing the effects of oxidative stress on the autophagy and apoptosis of GCs in broody geese.
Point 4: In line 122 “and T-AOC levels in serum during different periods (n = 8)” what do you mean “during different periods” in farm or in lab?
Response 4: For ease of description, “during different periods” meant to indicate “pre-broody period, laying period, and broody period”.
Point 5: In line 151 “form of apoptosis in the follicular GCs of Zhedong white geese in the laying period” is this the kind of geese used in the trail?
Response 5: We apologize for our unclear description. We wanted to express the morphological changes that are characteristic of apoptosis, such as chromatin margination, chromatin block scattering in the nucleus and highly compact chromatin block occupation.
Therefore, to avoid misunderstanding, we corrected the sentence as follows: “Apoptosis of GCs during the laying period, pre-broody period, and broody period observed by electron microscopy” (Lines 169-170)
Point 6: In line 195 - 196 “GCs were pretreated with 3-methyladenine (3-MA) to block autophagic activity prior to treatment with H2O2” is this a result?
Response 6: Yes, the description has been simplified for improve understanding. To verify the interactions between apoptosis and autophagy under H2O2-induced oxidative stress, we coincubated GCs with the autophagy inhibitor 3-MA, which prevents the formation and development of autophagosomes and thereby inhibits autophagy. Therefore, GCs were pretreated with 3-MA to block autophagic activity prior to treatment with H2O2. We confirmed that 3-MA could inhibit autophagy in GCs.
Point 7: What about prolactin and progesterone, which are responsible for the incubation of eggs? Are these two hormones related to the process of apoptosis and autophagy during broodiness?
Response 7: The main goal of our analysis was to investigate whether oxidative stress causes apoptosis and autophagy in broody goose. The relationships between prolactin, progesterone and apoptosis and autophagy during broodiness were not our primary focus. However, we agree with your point and will take it into account in our future studies.
Point 8: English should improve by a native person. The paper suffers from a poor English structure throughout and cannot be published or reviewed properly in the current format. The manuscript requires a thorough proofread by a native person whose first language is English.
Response 8: We apologize for our poor English and did not pay attention to details. To avoid language, grammar and spelling mistakes, the revised manuscript has been proofread and edited by American Journal Experts (www.aje.com/certificate; certificate verification key: 092D-43B6-12BB-B4BA-FFCP).
Point 9: The novelty of the study needs to be highlighted compared to other similar studies.
Response 9: Thank you for this suggestion. We have emphasized the innovation of this study in the Discussion (Line310-386) and Conclusion (Line 572-577) sections.
Point 10: Discussion is weak. The discussion needs enhancement with real explanations not only agreements and disagreements. Authors should improve it by the demonstration of biochemical/physiological causes of obtained results. Instead of justifying results, results should be interpreted, explained to appropriately elaborate inferences. discussion seems to be poor, didn't give good explanations of the results obtained. I think that it must be really improved. Where possible please discuss potential mechanisms behind your observations.
Response 10: We thank the reviewer for the helpful suggestion. The Discussion section has been rewritten. Some further explanations and meaning of the results have been added. More suggestions for future research have also been given. (Lines 310-386)
Point 11: You should also expand the links with prior publications in the area but try to be careful not to over-reach. For the latter, you should highlight potential areas of future study.
Response 11: Thank you. We have rewritten the manuscript as suggested.
Point 12: A detailed "Conclusion" should be provided to state the final result that the authors have reached.
Response 12: We have deleted this conclusion and rewritten the Conclusion according to the reviewer’s suggestion. (Conclusion section, Lines 572-577)
Overall, our results demonstrated serious oxidative stress, GC apoptosis and autophagy in broody geese, and oxidative stress promoted apoptosis and autophagy. Our study also provides the first line of evidence showing the importance of autophagy- and apoptosis-activated cell death in the attenuation of oxidative stress-induced goose broodiness using VE.
Point 13:Please note you only need to place your conclusion and not keep putting results, because these have already been presented in the manuscript.
Response 13: We re-wrote the conclusions according to the reviewer’s comment (Lines 572-577).
Point 14: Author(s) should reformat the references based on journal format. See the instructions for authors.
Response 14: We apologize for our carelessness. We have corrected and checked the entire paper.

Reviewer 2 Report
Dear Authour(s); The article offers important information for geese. The article offers important information for geese. The following information should be noted.
1. The necessary information about Vit E is not given in detail in the material and method section.
2. In the Summary and Conclusion section, Vit E is mentioned. However, more detailed information about vitamin E should be presented in the material method section. Because it's obviously an important factor. It should be emphasized and explained more. This situation should be clarified.
3. The colors and texts of the graphics are very difficult to read.
4. The image quality in the pictures is poor.
5. The number of animals used in the study and the number of samples should be given.
In general terms, the explanation and necessity of the study are well presented. The discussion is sufficient and explanatory. The presentation of the findings is good. But graphics and pictures should be rearranged (in terms of color and size). In the material method, more detailed information about vitamin E and the number of animals (geese) should be given.
Author Response
Responses to Reviewer Comments
Point 1: The necessary information about Vit E is not given in detail in the material and method section.
Response 1: Thank you for your advice. To elaborate on all the chemicals (containing vitamin E), we have revised the Materials and Methods section. Additional detailed information on the establishment of the oxidative stress model and antioxidant (containing vitamin E) treatment methods was also added.
Material and Methods
4.2 Chemicals (Lines 397-404)
MDC (catalog # 30432), hydrogen peroxide (H2O2; catalog # 323381), collagen (catalog # C7661), and type IV collagenase (catalog # C5138) were obtained from Sigma‒Aldrich, Inc. (St. Louis, MO, USA). A reactive oxygen species assay kit (DCFH-DA) was purchased from Beyotime Biotechnology (Shanghai, China). The autophagy inhibitor 3-MA (HY-19312) was purchased from MedChemExpress (NJ, USA). VE, proanthocyanins, resveratrol and gallic acid were purchased from Solarbio (Beijing, China).
4.7 Establishment of the oxidative stress model and antioxidant treatment (Lines 464-475)
GCs from goose ovaries were seeded into 96-well plates (1 x 104 cells/well). Once the cells reached 80%-90% confluence, the cells were treated with 0, 50, 100, 150, 200, or 250 μmol/L H2O2, and the plate was incubated at 38.5°C in 5% CO2 to induce oxidative stress-induced cell death. Twelve hours after treatment with H2O2, the cells were rinsed with warm PBS, and cell viability was assessed by CCK8 assay and LDH release assay. Based on a comprehensive assessment of the abovementioned test results, an optimal oxidative stress model was established. We then designed an antioxidant test for differential growth in the presence of optimal oxidative stress (100 μmol/L H2O2). Four different antioxidants, VE, proanthocyanins, resveratrol and gallic acid, were dissolved in DMSO (a solution of 0, 50, 100, 150, 200 and 250 μmol/L of each antioxidant was prepared) and separately added to GCs, and their antioxidant effects on H2O2-treated GCs were evaluated.
Point 2: In the Summary and Conclusion section, Vit E is mentioned. However, more detailed information about vitamin E should be presented in the material method section. Because it's obviously an important factor. It should be emphasized and explained more. This situation should be clarified.
Response 2: Thank you for this point. We have added related information to the Introduction and Materials and Methods sections. The revised text is indicated by “tracked changes” in the revised manuscript.
Introduction: We added some information on the mechanism or hypothesis of vitamin E (Lines 54-72).
Vitamin E (VE), curcumin and melatonin have been confirmed to exert protective effects against GC death [14,15,16].
Currently, the use of antioxidants has become a useful therapeutic approach against different livestock and poultry follicular atresia. VE is considered an ideal antioxidant because it can reach different body tissues depending on its lipophilic, anti-inflammatory, antioxidant and cytoprotective activities [17]. Some studies have clarified that VE plays a critical role in follicular development and growth and in follicular atresia by controlling proliferation, differentiation, and apoptosis [18,19]. Moreover, in recently published journal articles, Ayo JO et al. discussed the autophagic regulatory actions of VE supplementation in the performance of different livestock and poultry species, including rats, hens and reproductive cows [20,21,22].
The interaction between autophagy and apoptosis in cell death is very complex and unpredictable. Both mechanisms are involved in oxidative stress-induced cell death but can either antagonize or synergize with each other, and differences have been observed between different tissues of the same species and between the same tissues of different species. This finding encouraged us to investigate whether oxidative stress causes apoptosis and autophagy in broody geese and how it can be modulated by the antioxidant VE.
Material and Methods
The details are provided in our response to Point 1.
Point 3: The colors and texts of the graphics are very difficult to read.
Response 3: To address the reviewer’s concern, we have increased the font size of all the figures to improve their readability. Nonetheless, the color in the graphics remains poor because several graphics are a combination of multiple single graphics. To solve this problem, we have reprepared the image and enlarged the images of the necessary figures to ensure the clarity of each image. We hope that this change meets the requirements of the reviewer.
Point 4: The image quality in the pictures is poor.
Response 4: We apologize for our poor image quality. We have improved the quality of all figure images in the revised manuscript.
Point 5: The number of animals used in the study and the number of samples should be given.
Response 5: The number of animals and the groups of animals in the study were added to both the “Animals and sample collection” subsection of the Materials and Methods section, the supplementary material and the “figure notes” in the Results section of the manuscript.
The number of animals in the experimental group was selected from the viewpoint of animal welfare. We slaughtered and sampled 5 geese from each group (pre-broody period, laying period and broody period). In addition, we collected 8 blood samples to observe the oxidative stress levels in each group and used them for ELISA.
We provide a schematic of the experimental design below and in the manuscript. (Lines 578-581; Figure S).
Figure S. Experimental design and workflow for assessing the effects of oxidative stress on the autophagy and apoptosis of GCs in broody geese.

Reviewer 3 Report
The study determines the effect of oxidative stress on autophagy and apoptosis of granulosa cells in broody geese. The study lacks scientific language and appropriate flow of results. Additionally, there are numerous mistakes throughout the manuscript that need to be thoroughly checked. The study does not provide proper evidence to their claims and therefore needs to be reassessed after major corrections.
1- Introduction talks about mitophagy along with autophagy but does not provide details on it. Mitophagy involves numerous additional pathway and should either be explained or removed from the introduction.
- Please provide some background on the antioxidant “vitamin E” in introduction section.
- GL and TL has not been defined in the text. Additionally, in figure 1B, GL and TL (theca) measurement is not visible in broody period, please correct it.
- Figure 1E is very confusing, it shows cultured geese follicular GCs, but the result section mentions, it shows, decrease in GC no- There is no control to prove that.
- No proper labelling on figure 1F.
- Is there a reason why the H202 levels significantly come down in the broody period after pre-broody?
- How is figure 2E and F showing upregulation in nrf-1 and Ho-1? Please provide zoomed in images to be more clear and please label with arrows.
- What is the difference between figure D and G?
- Apoptotic molecules are expressed mostly translationally, please provide western blots to support your study in figure 3. Additionally, please provide western blots of procaspase-3 and its cleavage to caspase3, cytochrome c release to support the study that pre-broody and broody period shows apoptosis.
- No protein labelling is done in figure 3 C, G and I and therefore, it is very difficult understand the result.
- Immunohistochemistry analysis is not showing much difference in the levels of apoptotic proteins yet, the authors claim they do. Please provide zoomed images along with additional information to support your hypothesis.
- Figure 4E is not provided in the study. If the authors were referring figure 4f as 4e- How are the authors confirming that the green puncta are indeed autolysosomes? The pathway needs to be blocked in order to confirm it.
- What are a1, b1 and c1 in Figure 6B? are they the zoomed images of a, b, and c? In that case, visibility there are numerous dead cells in C1 in spite of adding vit E. Please acknowledge that and explain it.
- In figure 6D, there is no apparent decrease in ROS with vitamin E (visibly) yet the authors claim there is, please explain it
- No information of what figure is being referred to is done in result 2.7.
1- Numerous grammatical errors and typos all over the manuscript which needs to be corrected, few are mentioned below
2- Please remove “that are” after ROS in line no 48
3- Add “is” leading to in line no 58
4- The text in graphs in Figure 1 are not visible, please enlarge the texts.
5- Labelling of Figure 2 is very poor and no one of the labelled genes are visible to the reader. Please correct it.
6- Font variability in figure 6 E headings
7- Figure legends are inconsistent all over the manuscript.
Author Response
Responses to Reviewer Comments
Point 1: Introduction talks about mitophagy along with autophagy but does not provide details on it. Mitophagy involves numerous additional pathways and should either be explained or removed from the introduction.
Response 1: We thank the reviewer for the kind reminder. The term 'Mitophagy' has been removed in Line 50 of the Introduction.
Point 2: Please provide some background on the antioxidant “vitamin E” in introduction section.
Response 2: Thank you for this point. We have added the related information to the Introduction and Materials and Methods sections. The revised portions are indicated by “tracked changes” in the revised manuscript.
Introduction:
We added some information on the mechanism or hypothesis of vitamin E (Lines 54-72).
Vitamin E (VE), curcumin and melatonin have been confirmed to exert protective effects against GC death [14,15,16].
Currently, the use of antioxidants has become a useful therapeutic approach against different livestock and poultry follicular atresia. VE is considered an ideal antioxidant because it can reach different body tissues depending on its lipophilic, anti-inflammatory, antioxidant and cytoprotective activities [17]. Some studies have clarified that VE plays a critical role in follicular development and growth and in follicular atresia by controlling proliferation, differentiation, and apoptosis [18,19]. Moreover, in recently published journal articles, Ayo JO et al. discussed the autophagic regulatory actions of VE supplementation in the performance of different livestock and poultry species, including rats, hens and reproductive cows [20,21,22].
The interaction between autophagy and apoptosis in cell death is very complex and unpredictable. Both mechanisms are involved in oxidative stress-induced cell death but can either antagonize or synergize with each other, and differences have been observed between different tissues of the same species and between the same tissues of different species. This finding encouraged us to investigate whether oxidative stress causes apoptosis and autophagy in broody geese and how it can be modulated by the antioxidant VE.
Material and Methods: We have revised the Materials and Methods section. Additional detailed information on the establishment of the oxidative stress model and antioxidant (containing vitamin E) treatment methods was also added.
Material and Methods
4.2 Chemicals (Lines 397-404)
MDC (catalog # 30432), hydrogen peroxide (H2O2; catalog # 323381), collagen (catalog # C7661), and type IV collagenase (catalog # C5138) were obtained from Sigma‒Aldrich, Inc. (St. Louis, MO, USA). A reactive oxygen species assay kit (DCFH-DA) was purchased from Beyotime Biotechnology (Shanghai, China). The autophagy inhibitor 3-MA (HY-19312) was purchased from MedChemExpress (NJ, USA). VE, proanthocyanins, resveratrol and gallic acid were purchased from Solarbio (Beijing, China).
4.7 Establishment of the oxidative stress model and antioxidant treatment (Lines 464-475)
GCs from goose ovaries were seeded into 96-well plates (1 x 104 cells/well). Once the cells reached 80%-90% confluence, the cells were treated with 0, 50, 100, 150, 200, or 250 μmol/L H2O2, and the plate was incubated at 38.5°C in 5% CO2 to induce oxidative stress-induced cell death. Twelve hours after treatment with H2O2, the cells were rinsed with warm PBS, and cell viability was assessed by CCK8 assay and LDH release assay. Based on a comprehensive assessment of the abovementioned test results, an optimal oxidative stress model was established. We then designed an antioxidant test for differential growth in the presence of optimal oxidative stress (100 μmol/L H2O2). Four different antioxidants, VE, proanthocyanins, resveratrol and gallic acid, were dissolved in DMSO (a solution of 0, 50, 100, 150, 200 and 250 μmol/L of each antioxidant was prepared) and separately added to GCs, and their antioxidant effects on H2O2-treated GCs were evaluated.
Point 3: GL and TL has not been defined in the text. Additionally, in figure 1B, GL and TL (theca) measurement is not visible in broody period, please correct it.
Response 3: We greatly appreciate the reviewer’s careful comments regarding Lines 98-99 in the previous manuscript. In Lines 105 in the current manuscript, we define the abbreviations GL indicates the theca layer, TL indicates the granulosa layer. It is possible that the reviewer did not see these definitions due to the font size being too small.
Point 4: Figure 1E is very confusing, it shows cultured geese follicular GCs, but the result section mentions, it shows, decrease in GC no- There is no control to prove that.
Response 4: We apologize for our carelessness and have made changes based on the suggestions from the reviewer as follows.
“ We then assessed the potential development of GCs during broodiness in geese. Goose GCs were isolated and identified (Figure 2A, 2B), and the number of GCs was counted. The number of GCs was lower during the brooding period than during the laying period (P < 0.01 or P < 0.05) (Figure 2C).” (Lines 91-94)
Point 5: No proper labelling on figure 1F.
Response 5: Thank you for the suggestion. We have added proper labeling to the figure. The detailed correction is shown in Figure 2B. (Lines 109-110)
Point 6: Is there a reason why the H2O2 levels significantly come down in the broody period after pre-broody?
Response 6: Thank you for this question.
After checking the literature, we found no efficient study that demonstrates that the H2O2 levels significantly decrease in the brooding period. We apologize that we also did not find a suitable result during our research study and agree that this is a key underlying mystery that we will continue to explore.
Point 7: How is figure 2E and F showing upregulation in nrf-1 and Ho-1? Please provide zoomed in images to be more clear and please label with arrows.
Response 7: We apologize for our unclear description. Figures 2E and 2F, in the previous manuscript show only the results of immunohistochemical staining. To assess the Nrf2 and HO-1 expression levels, the mean density was assayed, which allowed better evaluation of the results (Figure 2G in the previous manuscript). We have recreated the image and enlarged the needed image to ensure appropriate clarity. We also included a brief explanation of the figure. (Lines 141-148)
Point 8: What is the difference between figure D and G?
Response 8: Figure 2D in the previous manuscript shows the relative expression of the Nrf2 and HO-1 genes as calculated by RT‒qPCR (Figure 4A in the revised manuscript). Figure 2G in the previous manuscript shows a quantitative analysis of the immunohistochemical staining of protein Nrf2 and HO-1 based on the mean density ( Figure 4C revised manuscript).
Point 9: Apoptotic molecules are expressed mostly translationally, please provide western blots to support your study in figure 3. Additionally, please provide western blots of procaspase-3 and its cleavage to caspase3, cytochrome c release to support the study that pre-broody and broody period shows apoptosis.
Response 9: Thank you for your good advice. Numerous studies have shown that immunohistochemistry can reveal protein expression levels [1,2,3,4,5,6]. Due to the members involved in the apoptosis pathway, antibodies are difficult to obtain for poultry. We have done our best to find the available resources and performed immunohistochemistry for Caspase-3. Unfortunately, no such antibodies, such as antibodies against pro-Caspase-3, cleaved Caspase-3, cytochrome c, Bcl-2, Caspase-8, Caspase-9 and p53, are available. Therefore, we performed RT‒qPCR to evaluate the expression of Caspase-3, Bcl-2, Caspase-8, Caspase-9 and p53 in the apoptosis signaling pathway during different periods.
Moreover, the search for available antibodies for the detection of apoptosis is still ongoing.
References:
[1] Zhu C, Zou C, Guan G, Guo Q, Yan Z, Liu T, Shen S, Xu X, Chen C, Lin Z, Cheng W, Wu A. Development and validation of an interferon signature predicting prognosis and treatment response for glioblastoma. Oncoimmunology. 2019 Jun 12;8(9):e1621677.
[2] Li Q, Liang J, Zhang S, An N, Xu L, Ye C. Overexpression of centromere protein K (CENPK) gene in Differentiated Thyroid Carcinoma promote cell Proliferation and Migration. Bioengineered. 2021 Dec;12(1):1299-1310.
[3] He X, Meng F, Qin L, Liu Z, Zhu X, Yu Z, Zheng Y. KLK11 suppresses cellular proliferation via inhibition of Wnt/β-catenin signaling pathway in esophageal squamous cell carcinoma. Am J Cancer Res. 2019 Oct 1;9(10):2264-2277.
[4] Bian ZQ, Luo Y, Guo F, Huang YZ, Zhong M, Cao H. Overexpressed ACP5 has prognostic value in colorectal cancer and promotes cell proliferation and tumorigenesis via FAK/PI3K/AKT signaling pathway. Am J Cancer Res. 2019 Jan 1;9(1):22-35.
[5] Song Y, Zhang C, Zhang J, Jiao Z, Dong N, Wang G, Wang Z, Wang L. Localized injection of miRNA-21-enriched extracellular vesicles effectively restores cardiac function after myocardial infarction. Theranostics. 2019 Apr 13;9(8):2346-2360.
[6] Tang C, Ni M, Xie S, Zhang Y, Zhang C, Ni Z, Chu C, Wu L, Zhou Y, Zhang Y. DICER1 regulates antibacterial function of epididymis by modulating transcription of β-defensins. J Mol Cell Biol. 2019 May 1;11(5):408-420.
Point 10: No protein labelling is done in figure 3 C, G and I and therefore, it is very difficult understand the result.
Response 10: We apologize for our unclear description. Figures 3C, 3G and 3F in the previous manuscript show the results of immunohistochemical staining. For assessment of the Caspase-3, LC3 and p62 expression levels, specific antibodies were used, and the average integrated optical density (IOD) of cells in 5 randomly chosen regions per sample was measured at a magnification of 200× using Image-Pro Plus 5.1 image analysis software. We have recreated the image and enlarged the image of the needed figure to ensure appropriate clarity (Figures 5C, 6C and 6D in the revised manuscript). We also rewrote a brief explanation of the figure. (Lines 167-188)
Point 11: Immunohistochemistry analysis is not showing much difference in the levels of apoptotic proteins yet; the authors claim they do. Please provide zoomed images along with additional information to support your hypothesis.
Response 11: We have corrected the issues in the section describing the immunohistochemistry analysis. The detailed corrections are detailed in the responses to Points 7~10.
Point 12: Figure 4E is not provided in the study. If the authors were referring figure 4f as 4e- How are the authors confirming that the green puncta are indeed autolysosomes? The pathway needs to be blocked in order to confirm it.
Response 12: We apologize for our mistake. We had mistakenly written ‘Figure 4E’ as ‘Figure 4F’. The point has been explicitly clarified and reordered.
Thank you very much for your objective evaluation of this research. As noted in the comment, the approach described in our paper explains why the green puncta are indeed autolysosomes. Please refer to '4.14 Monodansylcadaverine (MDC) staining for autophagic vacuoles' (Lines 566-571). In theory, we believe that green puncta can be considered autolysosomes.
4.14 MDC staining for autophagic vacuoles
MDC is a specific autophagy marker that can be absorbed by cells and selectively accumulate in autophagic vesicles. Foci of autophagic vesicles that are labeled with MDC and are distributed in the cytoplasm and around the nucleus can be observed by fluorescence microscopy. Therefore, the induction of autophagy can be determined based on changes in intracellular fluorescent particles [47].
References:
[47] Munafó, D.B. ; Colombo, M.I. A novel assay to study autophagy: regulation of autophagosome vacuole size by amino acid deprivation. J. Cell. Sci. 2001, 114, 3617–3629.
Point 13: What are a1, b1 and c1 in Figure 6B? are they the zoomed images of a, b, and c? In that case, visibility there are numerous dead cells in C1 in spite of adding vit E. Please acknowledge that and explain it.
Response 13: Yes, Figures 6B a1, b1 and c1 show zoomed images of a, b, and c. To avoid any possible misunderstanding, we have recreated the image in the manuscript.
In addition, thank you for this question. Please refer to lines 256-259: “First, the morphology of H2O2-treated GCs returned to normal after VE treatment (Figure 11B). Second, VE administration alleviated the H2O2-induced oxidative stress injury of the GC membrane, as determined by LDH release assay (Figure 11C).” We performed a potent antioxidant (vitamin E) rescue experiment but were unable to fully rescue the H2O2-reduced oxidative stress. Although numerous dead cells were detected, the degree of H2O2-induced cell death observed with vitamin E was lower than the degree of H2O2-mediated cell death. We hope our explanation satisfies the reviewer.
Point 14: In figure 6D, there is no apparent decrease in ROS with vitamin E (visibly) yet the authors claim there is, please explain it.
Response 14: Thank you for the kind advice. We have checked the indicated figure and quantified the ROS fluorescence intensity data in Figure 12A (Lines 271-277),right: ROS fluorescence intensity.
Figure 12. (A) Fluorescence probe staining of ROS observed under fluorescence microscopy at 488 nm and relative fluorescence intensity of ROS. Left: representative images of ROS staining; right: ROS fluorescence intensity.
Point 15: No information of what figure is being referred to is done in result 2.7.
Response 15: Thank you for your advice. We have added related figure information in subsection 2.7 of the Results section: “VE alleviates apoptotic and autophagic activities of H2O2-treated GCs.” (Lines 282-291)
Point 16: Numerous grammatical errors and typos all over the manuscript which needs to be corrected, few are mentioned below.
Response 16: We apologize for our poor English and did not pay attention to details. To avoid language, grammar and spelling mistakes, the revised manuscript has been proofread and edited by American Journal Experts (www.aje.com/certificate; certificate verification key: 092D-43B6-12BB-B4BA-FFCP).
Point 17: Please remove “that are” after ROS in line no 48.
Response 17: Thank you for the suggestion. We have deleted the phrase ‘that are’.
Point 18: Add “is” leading to in line no 58.
Response 18: We have corrected this mistake.
Point 19: The text in graphs in Figure 1 are not visible, please enlarge the texts
Response 19: Thank you for the objective evaluation of this research. We have increased the font size of all the figures to improve their readability.
Point 20: Labelling of Figure 2 is very poor and no one of the labelled genes are visible to the reader. Please correct it.
Response 20: We apologize for our poor image quality. We have improved the quality of all images in the figures in the revised manuscript.
Point 21: Font variability in figure 6 E headings.
Response 21: We have corrected the mistake.
Point 22: Figure legends are inconsistent all over the manuscript.
Response 22: We have corrected the mistake.

Round 2
Reviewer 3 Report
The authors have addressed all the questions
Author Response
Thank you again for your letter and the comments concerning our manuscript entitled “Effects of Oxidative Stress on the Autophagy and Apoptosis of Granulosa Cells in Broody Geese” (ijms-2108870).